# Relationship between Edinburg Postnatal Depression Scale (EPDS) Scores in the Early Postpartum Period and Related Stress Coping Characteristics

**DOI:** 10.3390/healthcare10071350

**Published:** 2022-07-21

**Authors:** Shoichi Magawa, Sachiko Yanase, Tsutako Miyazaki, Kazumasa Igura, Shintaro Maki, Shota Nii, Masafumi Nii, Hiroaki Tanaka, Eiji Kondo, Tomoaki Ikeda, Takayuki Kageyama

**Affiliations:** 1Department of Obstetrics and Gynecology, Faculty of Medicine, Mie University, Tsu-shi 5148507, Japan; mabochikin519@yahoo.co.jp (S.M.); irreg7@med.mie-u.ac.jp (S.N.); m-nii1984@med.mie-u.ac.jp (M.N.); h_tanaka@med.miyazaki-u.ac.jp (H.T.); eijikon@clin.medic.mie-u.ac.jp (E.K.); t-ikeda@clin.medic.mie-u.ac.jp (T.I.); 2Yanase Maternity Clinic, Tsu-shi 5140016, Japan; yanase-s@pmc.or.jp; 3Mie Prefectural College of Nursing, Tsu-shi 5140116, Japan; tsutako.miyazaki@mcn.ac.jp; 4Faculty of Nursing, Gifu Kyoritsu University, Oogaki-shi 5038550, Japan; igura@gku.ac.jp; 5Mental Health and Psychiatric Division, Oita University of Nursing and Health Sciences, Oita 8701163, Japan; kageyama@oita-nhs.ac.jp

**Keywords:** postpartum depression, Edinburgh Postnatal Depression Scale, Brief Scale for Coping Profile, primipara, multipara, gestational psychosis

## Abstract

Despite postpartum depression being a common mental health problem, there is no screening method for it. The only risk assessment used is the Edinburgh Postnatal Depression Scale (EPDS). We investigated the relationship between Brief Scale for Coping Profile (BSCP) subscales performed during pregnancy and EPDS scores. We recruited 353 women with normal pregnancies (160 primiparas, and 193 multiparas) and performed BSCP at 26 weeks of gestation. The EPDS was first performed within one week after delivery (T1), and then after one month (T2). Spearman’s correlation coefficients were calculated for the BSCP and EPDS for the whole and primi/multipara groups. Multiple regression analysis was performed with the EPDS T2 scores as the dependent variable. The EPDS scores were higher in the primipara group compared to the multipara (*p* < 0.001), and the EPDS T1 scores were higher than the overall T2 score (*p* < 0.001). In the multiple regression analysis, EPDS T1 and the “seeking help for solution” subscale were selected as significant explanatory variables when analyzed in the whole group; EPDS T1 and “active solution” for the primiparas; and EPDS T1, “changing mood”, and “seeking help for solution” for the multiparas. The BSCP can be used as a screening tool for postpartum depression during pregnancy.

## 1. Introduction

The prevalence of postpartum depression (PPD) is about 11% in women after childbirth [1]. However, there is a discrepancy in this rate among studies, which may be due to PPD being influenced by social and other factors. It has also been found that only 15% of postpartum women who experienced mood disorders in the first year after childbirth called for help or contacted a hospital [2]. Additionally, less than 50% of cases of PPD are detected by healthcare providers in routine practice [3]. Recently, however, PPD has received special attention, as suicide is the most common cause of death among postpartum mothers [4]. PPD is considered one of the most common mental health problems among women worldwide [5]. However, there is ample evidence that episodes of PPD are overlooked or misdiagnosed. Therefore, there is an urgent need for early detection and prevention. It is also important to identify the risk factors for PPD and predict its development. Several pregnancy risk factors have been identified, including low income, number of births, history of depression during pregnancy, history of mood disorders, abuse, and divorce [6,7,8,9,10,11,12].

The Edinburg Postnatal Depression Scale (EPDS) is widely used for screening PPD after childbirth [13]. However, the appropriate time to administer the EPDS and how the results change over time after pregnancy have not been elucidated. EPDS is a tool that can assess PPD in the postpartum period, and its use during pregnancy is limited. As a tool to assess depression during pregnancy, it has been suggested that the EPDS may be useful [14], but still there is no established method to assess the risk of PPD during pregnancy [14,15]. There is no consensus on a screening or risk assessment for PPD during normal pregnancies who do not have the obvious risk factors mentioned above. Implementation of predictive screening approaches by primary care providers before delivery is an important first step in PPD prevention and intervention efforts [16].

Life events such as pregnancy and childbirth are known to be important stressors [17], and an individual’s coping strategies are important, as they contribute to their stress responses [18]. Coping is defined as “a cognitive or behavioral effort to overcome, reduce, or accept external or internal pressures that are judged to be burdensome to the individual”. An individual’s coping profile, that is, “what coping strategies are usually chosen”, is known to be related to stress responses such as depressive symptoms in workers [19]. The relationship between such personal characteristics and PPD has also been reported [20,21]. The Brief Scale for Coping Profile (BSCP) has been developed as an assessment tool to measure stress coping characteristics, such as the coping profile of workers’ stressors [19,22]. Therefore, in the current study, the BSCP was used to evaluate the coping characteristics for stressors such as pregnancy, childbirth, and childcare in the near future, in a population with normal pregnancies. This study evaluated the usefulness of the BSCP as a predictive tool for PPD. The assessment of stress coping by the BSCP may help to explain why primipara mothers are a high-risk factor for PPD, as has been previously reported [23]. In other words, assessing the stress coping characteristics of primipara mothers versus multipara mothers may provide a solution to the question of why primipara mothers are at high risk and how they should be cared for.

Our study was conducted in a population with no history of psychiatric treatment during pregnancy and no abnormalities in the course of the pregnancy. In summary, we analyzed (1) the changes in the EPDS scores at discharge and when administered one month later; (2) the correlation between the BSCP subscales and the EPDS; and (3) the correlation between the factors of “primipara or multipara” during pregnancy, which was considered to be PPD risks, and the EPDS and BSCP items.

## 2. Materials and Methods

### 2.1. Study Design and Participants

This prospective observational study was conducted in a single hospital, from August 2019 to December 2020. Our study subjects were single pregnancy women of 18 years of age or older, in which no medical intervention for obvious fetal malformations or maternal complications (malformations, hypertension, diabetes, obesity, etc.) was required. The patients did not have a history of depression or other psychiatric disorders, nor did they have onset of illness during pregnancy. The study was conducted at a single hospital and included low-risk pregnancies. The recruited participants were 353 single pregnancy women. The participants completed all the questionnaires required at 26 weeks of gestation (T0), within one week of delivery (T1), and in the first postpartum month (T2). All women took the BSCP [19] and questionnaires on whether they were primipara or multipara at T0 (Appendix). In addition, the EPDS was performed at T1 and T2. All participants provided written informed consent prior to participation in the study.

### 2.2. Materials

*EPDS:* The EPDS, a 10-item self-report measure, was developed specifically to be used with a community sample of postpartum mothers [13]. Responses were rated on a 4-point scale (scored from 0 to 3) and indicated the extent to which each statement corresponded to their mood over the past seven days. The sum of the item scores was the total score, and higher scores indicated more depressive symptoms. The EPDS has been validated and shows adequate concurrent validity, with a standardized Cronbach’s alpha of 0.82 [24]. For Japanese postpartum women, a cutoff point of nine was considered optimal for the screening of PDD [25]. Studies on Japanese women have confirmed that the EPDS has good psychometric properties and is a valid and reliable screening tool [25,26].

*BSCP:* The BSCP consisted of 18 items and was rated on a 4-point scale. It had six subscales: “active solution”, “seeking help for a solution”, “changing mood”, “avoidance and suppression”, “changing a point of view”, and “emotional expression involving others” that assessed the tendency to select a coping profile for stress [19,22]. Stress coping can be divided into problem-centered coping characteristics and emotion-centered coping characteristics. The former consists of “active solution” and “seeking help for a solution”, and the latter consists of “changing mood”, “avoidance and suppression”, “changing a point of view”, and “emotional expression involving others”. If a respondent scored high on a subscale, it meant that she frequently chose such a strategy for coping with stress.

### 2.3. Statistical Methods

Spearman’s correlation coefficient was calculated to determine the correlation between the BSCP subscale scores and the EPDS scores. Changes in EPDS scores within one week and one month after delivery were tested by using a paired *t*-test. In addition, for the association between the mothers, the results were divided into primipara and multipara mothers. Multiple regression analysis was performed with the EPDS scores at one month postpartum as the dependent variable and the other variables as the independent variables. This analysis was performed for all cases across the primipara and multipara groups. The mean ± standard deviation was used for notation, and the significance level was set at *p* = 0.05. All statistical analyses were performed by using SPSS 26.0 (version 26; IBM Corporation, Armonk, NY, USA).

## 3. Results

### 3.1. Reliability of the Questionnaire

The Cronbach’s alpha coefficients for the BSCP subscales ranged from 0.63 to 0.86 (0.72 for “active solution”, 0.71 for “seeking help for a solution”, 0.81 for “changing mood”, 0.71 for “avoidance and suppression”, 0.86 for “changing a point of view”, and 0.63 for “emotional expression involving others”.

### 3.2. Changes in the EPDS at Discharge and One Month after Delivery

Compared to the EPDS scores at discharge (M = 3.55; SD = 3.65), the scores after one month (M = 2.77; SD = 2.95) were significantly lower, *p* < 0.001. There was a positive correlation between the EPDS scores at discharge and the scores after one month (r = 0.509; *p* < 0.001), and the number (%) of patients with a score of nine or higher decreased from 23 (6.5%) to 13 (3.7%) (Table 1).

### 3.3. Correlation between the BSCP Subscales and EPDS at Discharge and One Month after Delivery

Among the subscales of the BSCP, “changing mood” showed a negative correlation with the EPDS scores at discharge. In addition, all three scores of “active solution”, “seeking help for a solution”, and “changing mood” were negatively correlated with the EPDS scores after one month postpartum (Table 2).

### 3.4. Correlations and Changes between First-Time/Previous Mothers and the BSCP and EPDS Scores

The target population was divided into two groups: first-time mothers (primipara group) and multipara women (multipara group). There were 160 women in the primipara group and 193 in the other. The EPDS scores at hospital discharge were significantly higher in the primipara group compared to those for the multipara group (4.55 ± 3.91 vs. 2.70 ± 3.22, *p* < 0.001). The results of EPDS scores at one month showed significantly higher scores for the primipara group compared to the multipara group (3.41 ± 2.99 vs. 2.24 ± 2.82, *p* < 0.001).

In the BSCP, the “avoidance and suppression” score was higher in the multipara group than in the primipara group (6.21 ± 1.97 vs. 6.71 ± 2.24, *p* = 0.029). There were no differences in the other BSCP subscale scores between the two groups.

A multiple regression analysis was performed with the EPDS scores at one month postpartum as the objective variable. When all the participants were analyzed, the EPDS at discharge and “seeking help for a solution” subscale were selected as significant explanatory variables. The results were also analyzed separately for the primipara or multipara group. The results indicated that EPDS at discharge and the “active solution” subscale were selected as explanatory variables for the primipara group. Moreover, for the EPDS at discharge, the “changing mood” and “seeking help for a solution” subscales were selected as explanatory variables for the multipara group (Table 3).

## 4. Discussion

In this study, we observed the following:The EPDS scores at one month were positively correlated with and significantly lower than the EPDS scores at discharge. Additionally, the percentage of high-scorers also decreased;Primipara mothers had significantly higher EPDS scores at discharge and after one month compared to multipara mothers.Multiple regression analysis using the EPDS score at one month as the objective variable showed that the EPDS scores at hospital discharge were a predictor of the EPDS score at one month. EPDS scores at hospital discharge also predicted EPDS scores at one month when the participants were divided between primipara and multipara mothers.

Additionally, the “seeking help for a solution” subscale overall, “active solution” in the primipara group, and “changing mood” and “seeking help for a solution” in the multipara group were also independent predictors of the EPDS scores at one month.

Previous reports have shown that women with PPD tended to experience a decrease in symptoms and EPDS scores over time without treatment [27,28], as is consistent with our results. However, no studies have evaluated the changes in the short postpartum period, but the current study found that EPDS scores changed even within a short period of one month after delivery. This was consistent with previous reports that the incidence of PPD decreased with time after delivery [6,29]. In this study, we assessed the BSCP subscale at 26 weeks of gestation and the EPDS at hospital discharge and one month postpartum. Previous studies on primipara mothers have found that they were more anxious about childcare and had lower self-esteem and stress compared to multipara women [30,31]. Since previous reports have also indicated that primipara mothers were a risk factor for PDD, and EPDS scores were higher in primipara mothers than in multipara mothers [23,32], we conducted subgroup analyses. In the present study, primipara mothers also had higher EPDS scores than multipara mothers, as is consistent with previous reports.

Among the coping characteristics assessed after 26 weeks of gestation, the “active solution” score for primipara mothers and the “seeking help for a solution” score for multipara mothers showed a negative correlation to the EPDS scores at one month postpartum when statistically adjusted for the effect of EDPS score at discharge. Together, these two scores indicated problem-centered coping. Thus, the results suggested that both primipara and multipara mothers were more likely to be depressed in the first postpartum month if they had not developed problem-centered coping characteristics [18], or it was suggested that the characteristics of problem-centered coping may have a protective effect against PPD. In primipara mothers, it is conceivable that those who actively confront the new task of postpartum childcare and repeat trial-and-error methods may increase their self-affirmation by accumulating successful experiences, which may result in a decrease in the EPDS scores. It has been reported that a decrease in self-affirmation in the early postpartum period improved as the postpartum months passed, and the EPDS score decreased accordingly [33]. Multidisciplinary maternal and child health services in the community are important to support pregnant women with less problem-centered coping characteristics [34].

Although there are few studies on PPD that focus only on multipara mothers, according to a large cohort study in Japan, where multipara mothers made up about half of the sample, cohabitation with a partner and their participation in childcare during the first postpartum month were associated with lower EPDS scores [35]. In addition, support from family and partners has been reported to be important in improving the quality of life of pregnant women [36]. Multipara mothers, in contrast, have experienced at least one postpartum problem and may know how to cope with their life needs and when to seek help from others. However, the dual task of caring for a second child along with caring for the first child is still new. Even if they have acquired the coping skill of “active solution”, they may need help from others, such as their partner, to carry out the dual task. Therefore, the coping trait of seeking help from others instead of taking on the dual task alone is thought to be useful in reducing depression.

In addition to problem-centered coping, frequent use of “changing mood”, a type of emotion-centered coping, also contributed to lower EPDS scores in the first postpartum month. Considering these previous studies and the results of the present study, a conclusion can be made. Although multipara mothers are knowledgeable on how to solve various challenges during the postpartum period, using emotion-centered coping, such as mood changes, is more effective in reducing the stress response (strain), while also utilizing human resources such as partners, rather than solving the problem alone. It is thought that the flexibility to use these two coping characteristics in different ways reduces depression. While there are many ways to “change one’s mood” (e.g., talking to a close friend), it requires a certain amount of time, money, partnership, and other costs. Therefore, it is important to have these resources. To test this conjecture, future studies should examine the specific methods of mood change that are strongly associated with lower EPDS scores in the postpartum period. In contrast, for primipara mothers, there are many more problems to be solved through problem-centered coping during the postpartum period than through emotion-centered coping. Therefore, we suspect that the relationship between the “changing mood” score and the EPDS score was not found.

Although various interventions have been tried to prevent PPD, none has been found to be effective in the general population [37]. In contrast, there are some reports suggesting that PPD can be prevented by limiting the target population to high-risk cases [38,39]. The causes of PPD are varied and require a multifaceted approach. In this study, we investigated the possibility of preventing PPD in a low-risk pregnant population. We evaluated the risk of developing PPD by assessing whether it was the primipara or not, in addition to the perspective of stress coping. In the future, appropriate intervention methods must be considered. The results of this study suggest that stress coping that is a risk factor for PPD differs between primipara mothers and multipara mothers. We need to understand these characteristics and develop intervention methods to obtain appropriate stress coping.

### Limitations

One limitation of this study was that the sample was limited to patients from a single primary center. Future research should consider how the relationship between the BSCP subscale and the EPDS may change in a group of patients considered to be at higher risk. Moreover, we do not collect demographic data such as the income or educational background of eligible patients.

## 5. Conclusions

This study found that the scores on the BSCP subscale items administered during pregnancy were associated with the EPDS scores in the postpartum period. In particular, there was a close relationship between the EPDS scores one month after childbirth and problem-centered coping at 26 weeks of gestation. Moreover, there was a difference in the coping characteristics that contributed to postpartum depressive tendencies between primipara and multipara mothers. IT may be useful to apply BSCP during pregnancy to design effective preventive intervention and support measures for PPD. Future research should examine whether it is possible to prevent the onset of PPD by examining screening methods and individualized preventive interventions for risk factors of postpartum depression during pregnancy.

## Figures and Tables

**Table 1 healthcare-10-01350-t001:** Correlation with changes in EPDS at discharge and at 1 month of discharge.

	Time of Discharge	One Month after Discharge	*p*	Pearson’s Correlation Coefficient
	Mean ± SD	Mean ± SD		
EPDS	3.55 ± 3.65	2.77 ± 2.95	<0.001 ***	0.509 ***

*** *p* < 0.001.

**Table 2 healthcare-10-01350-t002:** Correlation between BSCP subscales collected at 26 weeks gestation and EPDS (at hospital discharge and 1 month after discharge).

		EPDS Score	
		Time of Discharge	One Month after Discharge
Active solution	coefficient	−0.051	−0.135 *
	*p*	0.335	0.011
Seeking help for solution	coefficient	−0.078	−0.201 **
	*p*	0.143	<0.01
Changing mood	coefficient	−0.127 *	−0.185 **
	*p*	0.017	<0.01
Avoidance and suppression	coefficient	−0.039	−0.065
	*p*	0.460	0.225
Changing a point of view	coefficient	0.017	−0.036
	*p*	0.752	0.496
Emotional expression involving others	coefficient	0.076	0.027
	*p*	0.156	0.617

** < 0.01; * < 0.05.

**Table 3 healthcare-10-01350-t003:** Multiple regression analysis with EPDS score at 1 month postpartum as the objective variable.

Factors Associated with EPDS 1 Month (Overall)		
	β	*p*
EPDS score(discharge)	0.399	<0.001
Seeking help for solution	−0.257	<0.001
F value	68.038	
R^2^	0.284	
adjusted R^2^	0.280	
Factors associated with EPDS 1 month (primipara)		
	β	*p*
EPDS score(discharge)	0.334	<0.001
Active solution	−0.297	0.026
F value	22.128	
R^2^	0.222	
adjusted R^2^	0.212	
Factors associated with EPDS 1 month (multipara)		
	β	*p*
EPDS score(discharge)	0.401	<0.001
Seeking help for solution	−0.293	0.001
Changing mood	−0.211	0.005
F value	35.147	
R^2^	0.364	
adjusted R^2^	0.354	

## Data Availability

Not applicable.

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
