# Peer review of "Relationship between Edinburg Postnatal Depression Scale (EPDS) Scores in the Early Postpartum Period and Related Stress Coping Characteristics"

_healthcare, 2022, doi:10.3390/healthcare10071350_

Round 1
Reviewer 1 Report
The manuscript is well-written and interesting, and examining the relationship between Brief Scale for Coping Profile (BSCP) subscales performed during pregnancy and EPDS scores. A comprehensive study with 353 women with normal pregnancies participants was performed to address this question. However, there are still some minor revisions that need to be done. If they can be done, the manuscript could make more contribution. After a careful review of this manuscript, the reviewer has the following comments:
1.The authors should clearly state the informed written consent in the research methods section.
2.The number of decimal places in this article is different from that in the table, and it is suggested that they should be unified, eg. In Line 123 SD=3.65, in Table 1 SD=3.653, in Line 123 SD=2.95, in table 1 SD=2.949.
3. In table 3, R2 an adjusted R2 should be R2 and adjusted R2, the F score should be F value.
Author Response
Review1
The manuscript is well-written and interesting, and examining the relationship between Brief Scale for Coping Profile (BSCP) subscales performed during pregnancy and EPDS scores. A comprehensive study with 353 women with normal pregnancies participants was performed to address this question. However, there are still some minor revisions that need to be done. If they can be done, the manuscript could make more contribution. After a careful review of this manuscript, the reviewer has the following comments:
- The authors should clearly state the informed written consent in the research methods section.
Thank you for pointing this out, we added the following text in Methods section.
[All participants provided written informed consent prior to participation in the study.]
- The number of decimal places in this article is different from that in the table, and it is suggested that they should be unified, eg. In Line 123 SD=3.65, in Table 1 SD=3.653, in Line 123 SD=2.95, in table 1 SD=2.949.
I have corrected it as you indicated.
- In table 3, R2 an adjusted R2 should beR2and adjusted R2, the F score should be F value.
I have corrected it as you indicated.

Reviewer 2 Report
This study aimed to assess the relationship between the Brief Scale for Coping Profile (BSCP) and Edinburgh Postnatal Depression Scale (EPDS) for post-partum depression screening, changes in EPDS scores over time, and the association of being primiparous or multiparous and EPDS and BSCP items.
Introduction: The first sentence in the 2nd paragraph states that “EPDS is widely used for screening PPD after childbirth”, yet lines 47 and 48 seemingly contradict this by stating that “EPDS is useful in screening for depression during pregnancy, it is not clear whether the same results are also useful in screening for PPD”. This paragraph would benefit from clearly stating what the current and supported usages of the EPDS is, as well as the usages we are unclear about and hope to elucidate.
There should be some discussion on the most current literature in this area, perhaps mention of other studies conducted on predictive screening for PPD. It would be beneficial to explicitly state the current knowledge gap and the aim of this study. Is the study attempting to identify predictive factors for PPD? Assess the utility of the BSCP as a predictive tool for PPD?
The introduction states that “this study further aims to predict high-risk groups for postpartum depression during pregnancy” (line 65-66), in addition to assessing “the association with factors such as primipara”. However, the rest of the study does not appear to address risks in particular groups beyond being a primiparous vs multiparous mother.
Methods: There is a confusing description of the study sample. Line 79 identifies “normal singleton pregnant women” as the study population. There is no standard for what “normal” means, and “singleton” is ambiguous, often informally used to indicate someone whose relationship status is single. There should also be a better description of the population this study aims to address targets, and how reflective this sample is of the population.
Justification for not collecting demographic data such as income was “to ensure patient confidentiality” (line 82). However, without this data and especially in a study assessing predictive factors, this makes the study liable to confounders. Confidentiality can be assured while still collecting information such as education and socioeconomic status, which may be associated with both coping strategies and depressive symptoms. A better justification is required for why demographic factors would not conceivably confound results.
There is inadequate explanation for what the BSCP scale items mean. This is later explained in the discussion section, but should be explained in the methods section.
Results: This section is well organized. Lines 146 and 147 are worded in a way that is confusing and apparently contradictory. It appears to simultaneously state that the EPDS scores between the two groups were similar to each other, but also that one was significantly higher than the other.
Discussion: Definitive statements such as “problem-centred coping characteristics had a protective effect against PPD” should be avoided. There was no discussion of potential confounders in this study, and thus causation and protective effects cannot be ascertained. Lack of demographic data should be addressed in limitations.
There should be more in-depth discussion of future research as well as implications for policy and practice in the discussion section, beyond a few sentences in the conclusion.
Conclusion: The study is well summarized.
Author Response
Review2
This study aimed to assess the relationship between the Brief Scale for Coping Profile (BSCP) and Edinburgh Postnatal Depression Scale (EPDS) for post-partum depression screening, changes in EPDS scores over time, and the association of being primiparous or multiparous and EPDS and BSCP items.
Introduction: The first sentence in the 2nd paragraph states that “EPDS is widely used for screening PPD after childbirth”, yet lines 47 and 48 seemingly contradict this by stating that “EPDS is useful in screening for depression during pregnancy, it is not clear whether the same results are also useful in screening for PPD”. This paragraph would benefit from clearly stating what the current and supported usages of the EPDS is, as well as the usages we are unclear about and hope to elucidate.
Thank you for pointing this out, the EPDS is essentially a tool that is assessed in the postpartum period and assesses PPD. Some studies have attempted to assess depression during pregnancy by using an extended version of the EPDS and assessing it during pregnancy.
However, the results of this study were that it does not assess PPD, which is postpartum psychosis as it is. As you noted, the text is difficult to follow and has been changed as follows.
In addition, we thought it necessary to describe predictive screening in terms of previous reports, results, and methodological aspects of medical intervention, and added it to the Discussion section because of the increased amount of text.
[The EPDS is a tool that can assess PPD in the postpartum period, and its use during pregnancy is limited. As a tool to assess depression during pregnancy, it has been suggested that the EPDS may be useful [14], but still there is no established method to assess the risk of PPD during pregnancy [14,15].]
There should be some discussion on the most current literature in this area, perhaps mention of other studies conducted on predictive screening for PPD. It would be beneficial to explicitly state the current knowledge gap and the aim of this study. Is the study attempting to identify predictive factors for PPD? Assess the utility of the BSCP as a predictive tool for PPD? The introduction states that “this study further aims to predict high-risk groups for postpartum depression during pregnancy” (line 65-66), in addition to assessing “the association with factors such as primipara”. However, the rest of the study does not appear to address risks in particular groups beyond being a primiparous vs multiparous mother.
As you indicated, the primary goal of this study was to ascertain whether BSCP during pregnancy can be used to assess PPD risk. During this process, we found that the results were different for primipara mothers and multipara mothers, and we arrived at the conclusion that we need to consider how to intervene in their care. We have changed the description as follows.
[This study evaluated the usefulness of the BSCP as a predictive tool for PPD. The assessment of stress coping by the BSCP may help to explain why primipara mothers are a high-risk factor for PPD, as has been previously reported. In other words, assessing the stress coping characteristics of primipara mothers versus multipara mothers may provide a solution to the question of why primipara mothers are at high risk and how they should be cared for.]
Methods: There is a confusing description of the study sample. Line 79 identifies “normal singleton pregnant women” as the study population. There is no standard for what “normal” means, and “singleton” is ambiguous, often informally used to indicate someone whose relationship status is single. There should also be a better description of the population this study aims to address targets, and how reflective this sample is of the population.
Thank you for your suggestions. First, we have changed the description of singleton to single pregnancy. In addition, since the phrase "normal pregnancy" is very ambiguous, we have added the following statement: [Our study subjects were single pregnancy women of 18 years of age or older, in which no medical intervention for obvious fetal malformations or maternal complications (malformations, hypertension, diabetes, obesity, etc.) was required. The patients did not have a history of depression or other psychiatric disorders, nor did they have onset of illness during pregnancy. The study was conducted at a single hospital and included low-risk pregnancies.]
Justification for not collecting demographic data such as income was “to ensure patient confidentiality” (line 82). However, without this data and especially in a study assessing predictive factors, this makes the study liable to confounders. Confidentiality can be assured while still collecting information such as education and socioeconomic status, which may be associated with both coping strategies and depressive symptoms. A better justification is required for why demographic factors would not conceivably confound results.
In Western countries, it is possible to estimate income to some extent based on insurance coverage, and this method has actually been used to evaluate the relationship between income, educational background, and PPD. In Japan, however, the national health insurance system is universal, so it is difficult to estimate income, and we did not assume that we would ask about income and educational background at the research planning stage.
The related section has been deleted.
In the Limitation section, we added [We do not collect demographic data such as income or educational background of eligible patients.]
There is inadequate explanation for what the BSCP scale items mean. This is later explained in the discussion section, but should be explained in the methods section.
Thank you for your suggestion, I have added the following statement in the Methods section for better understanding.
[Stress coping can be divided into problem-centered coping characteristics and emotion-centered coping characteristics. The former is "Active solution," "Seeking help for a solution," and the latter is "Changing mood," "Avoidance and suppression," "Changing a point of view," and "Emotional expression involving others.]
Results: This section is well organized. Lines 146 and 147 are worded in a way that is confusing and apparently contradictory. It appears to simultaneously state that the EPDS scores between the two groups were similar to each other, but also that one was significantly higher than the other.
We have removed the word "similar".
Discussion: Definitive statements such as “problem-centred coping characteristics had a protective effect against PPD” should be avoided. There was no discussion of potential confounders in this study, and thus causation and protective effects cannot be ascertained.
Thank you for pointing this out.
We have limited the statement to "It was suggested that the characteristics of problem-centered coping may have a protective effect against PPD."
Lack of demographic data should be addressed in limitations.
As mentioned above, we have added a description to the Limitation section.
There should be more in-depth discussion of future research as well as implications for policy and practice in the discussion section, beyond a few sentences in the conclusion.
As you suggested in the Introduction section, we have added a statement based on the results of previous studies and issues. We have added the following statement based on the results and challenges of previous studies:
[Although various interventions have been tried to prevent PPD, none have been found to be effective in the general population [37]. In contrast, there are some reports suggesting that PPD can be prevented by limiting the target population to high-risk cases [38,39]. The causes of PPD are varied and require a multifaceted approach. In this study, we investigated the possibility of preventing PPD in a low-risk pregnant population. We evaluated the risk of developing PPD by assessing whether it was the primipara or not, in addition to the perspective of stress coping. In the future, appropriate intervention methods must be considered. The results of this study suggest that stress coping that is a risk factor for PPD differs between primipara mothers and multipara mothers. We need to understand these characteristics and develop intervention methods to obtain appropriate stress coping.]
